# Differential Seasonality of Legionnaires’ Disease by Exposure Category

**DOI:** 10.3390/ijerph17093049

**Published:** 2020-04-28

**Authors:** Udo Buchholz, Doris Altmann, Bonita Brodhun

**Affiliations:** Department for Infectious Disease Epidemiology, Robert Koch Institute, Seestraße 10, 13353 Berlin, Germany; AltmannD@rki.de (D.A.); BrodhunB@rki.de (B.B.)

**Keywords:** Legionnaires’ disease, surveillance, seasonality

## Abstract

Legionnaires’ disease (LD) shows a seasonal pattern with most cases occurring in summer. We investigate if seasonality can be observed for all three exposure categories (community-acquired (CALD), travel-associated (TALD) and healthcare-associated (HCA)). Methods: LD cases (2005–2015) were classified by exposure categories and we calculated the relative case ratio for each month from February to December using January as reference. The TALD relative case ratio was compared with flight frequencies. Results: Overall case numbers in Germany (N = 7351) peaked in August. CALD had a curve similar to all cases. TALD displayed a bimodal curve with peaks in June/July and October. The latter was attributable to LD cases aged 60+. The relative case ratio of TALD surpassed clearly that of CALD. The curve was similar to that of the relative flight frequencies, but was shifted by about one month. HCA showed no apparent seasonality. Conclusions: Although the overall seasonality in LD is heavily influenced by CALD, seasonal differences are more pronounced for TALD which may reflect travel behavior. The bimodal pattern of TALD is attributable to the curve among those aged 60+ and may reflect their preference to travel outside school holiday periods. Heightened vigilance for HCA cases is necessary throughout the entire year.

## 1. Introduction

The incidence of Legionnaires’ disease (LD) follows a seasonality with more cases in the summer and fall [1,2]. According to exposure, cases of LD are divided into three exposure categories: (1) community-acquired LD (CALD), (2) travel-associated LD (TALD) and (3) healthcare-associated LD (HCA). It is unclear if one or all of the exposure categories follow a seasonal pattern and how these could be explained. We aimed to explore pooled German and—for comparison—European surveillance data to answer the above questions.

## 2. Materials and Methods

In Germany, the Infection Protection Act (Infektionsschutzgesetz (IfSG)) requires that LD is reportable since 2001 [3]. Local health departments record—among other variables—month and year of birth, sex, date of illness onset, presence of pneumonia and exposure before infection. Exposure to a hospital, rehabilitation center or nursing home is grouped as HCA. Locally recorded data are transmitted in an anonymized form via state health departments to the Robert Koch Institute. Cases who were not categorized as HCA or TALD and whose location of infection was the same as the district who reported the case, were assumed to be CALD.

We pooled data of cases notified from 2005 to 2015 in Germany. The overall number and the exposure-category-specific number of cases were displayed by month of illness onset. To better express the degree of seasonality by exposure category, we calculated the ratio of LD cases for each month and used January of the same year as reference. We intended to examine if epidemiologic patterns depend on demographic parameters. For example, retired persons may have different travel preferences than families. For exploratory purposes we stratified the relative case ratio of the exposure categories by age group (<60 years and 60+).

In Europe, an agreement exists to a set of infectious diseases that are annually notifiable to the European Center for Disease Prevention and Control (ECDC). The database where these data are uploaded to is named “The European Surveillance System” (TESSy). It is possible for individuals nominated by the EU/EEA countries, EU commission, EU bodies, international organizations and other entities to submit requests for subsets of TESSy data. We requested aggregated data on age group (<60 years and 60+), exposure category (CALD, TALD, Hospital) as well as year and month of illness onset. The dataset provided by ECDC included 50,284 European cases for the years 2005–2015.

In Germany the four countries Spain, Italy, Greece and Turkey are reported in more than 50% of travel destinations among cases with TALD. For Germany the number of flights to these four countries was available for the years 2011–2015 [4]. We compared the relative case ratio of TALD cases from 2011 to 2015 (pooled for the years; using January as reference) with the relative number of flights to the countries Spain, Italy, Greece and Turkey (pooled for the years and countries; using January as reference).

We performed data analysis using Microsoft Excel (Microsoft Office Professional Plus 2010, Redmond, WA, USA).

Ethics statement: As a federal law, the German Infection Protection Act (Infektionsschutzgesetz (IfSG)) regulates the prevention and management of infectious diseases in humans. In order to guarantee confidentiality, the IfSG requires that notified data are reported anonymously to the national authority.

## 3. Results

We included 7351 LD cases reported between 2005 and 2015 in the analysis of the German Data. Most cases were attributed to CALD (5415 cases; 74%) followed by TALD (1477 cases; 20%) and HCA (459 cases; 6%).

The pooled overall case number by month showed a clear peak in August (1030 of 7351 cases; 14%; Figure 1 left panel). Numbers of CALD cases followed a similar curve as that of all cases. The relative case ratio of CALD cases reached 1.8 (Figure 1 right panel).

The curve of HCA cases (red line in Figure 1 left panel) showed no apparent seasonality, whereas the curve of CALD cases shows a clear seasonal pattern with a peak in the summer months. The curve of TALD cases rather plateaus during summer months, or even shows a dent when being “amplified” in the curve showing the relative case numbers (green line in Figure 1 right panel). The relative case ratio of TALD cases surpassed that of the CALD cases and reached 2.9 in July and 3.8 in October (green line in Figure 1 right panel). Moreover, the relative case ratio of TALD cases rose already in the months March–May, while the relative case ratio of CALD cases (blue line) exceeded the January value of 1 only from June–October. After stratification into age groups <60 years and 60+, the two CALD curves agreed almost exactly with the seasonal course. (Figure 2 left panel). The two TALD curves differed: the curve of those aged <60 years had a shorter “season” from June until October and a unimodal course with a broad peak in August and September, the curve of those aged 60 year or more showed a bimodal course with peaks in June/July and an even higher peak in October (Figure 2 left panel). Relative case ratios emphasize these observations even more. In addition, it can be seen more clearly that the “season” of those aged 60 years and above lasts from April until November with a drop of cases in August, but that among those aged <60 years lasts only from June until October (Figure 2 right panel). Data of other countries reporting to ECDC “behave” very similarly (Figure 3 and Figure 4).

Comparisons of the monthly relative case ratio of TALD cases from 2011 to 2015 with that of the number of flights to Spain, Italy, Greece and Turkey shows a good agreement (Figure 5). There is a slight shift of the flight frequency curve to that of TALD cases of approximately one month. The October peak among TALD cases (due to those aged 60+ years; Figure 2 right panel) is not reflected in the curve of the flight frequency.

## 4. Discussion

We have used two simple analyses to reveal a couple of interesting properties of LD. Using relative case ratios (instead of absolute numbers) show that the degree of seasonality is much stronger for TALD than for CALD; moreover the annual rise starts earlier for TALD than for CALD cases, and this is mainly due to the case load among those 60+ years old. Nevertheless, the seasonal burden of disease is higher for CALD cases. The summer rise of CALD is not entirely understood but (among other reasons) may have to do with increased temperature in drinking water installations.

It has been estimated that the absolute risk (per person-time of travelling) of acquiring TALD is almost 10 times higher compared to acquiring CALD [5]. This and the stronger seasonality observed here may be due to a double risk of travelers: (a) exposure to hotels or other accommodations which often have complex or old plumbing systems and are not always fully booked, and (b) travelling is often associated with stagnant drinking water in the home of the traveler during the period of travelling. Stagnation of drinking water in pipes and outlets might lead to proliferation of Legionellae posing a potential risk of acquiring LD upon return. Although this mode of acquisition of LD should be really counted as CALD, in practice these two origins of acquiring LD cannot usually be differentiated by the health department in charge. It is also unclear which proportion of TALD is attributable to which of the named risks above. In addition, summer tourism takes place when the risk of contracting LD (in the traveled countries) is generally higher due to higher temperatures and possibly better growth conditions for Legionella in the natural environment or cooling towers. Thus, this fact may also play a role in the seasonality of TALD.

Although based on an “ecological” analysis, it is interesting to see that the shape of the curve of the relative TALD case count coincides with that of the relative flight frequency to the four countries that comprise more than half of all destinations named among TALD cases in Germany (Spain, Italy, Greece, Turkey). Of course, flight frequency can only be used as a proxy for all travels to these (and other) countries. The observed shift might be partially explained by the time needed to be infected at the travel destination and the 2–10 day incubation period. The two curves differ only for the months of September and October when the TALD curve actually reaches its peak whereas the flight frequency curve is already on its decline after a peak in July. The TALD peak is due to the second peak among those aged 60+ years (Figure 2 right panel) which again is related to travels to the four countries of Spain, Italy, Greece and Turkey (data not shown). Deeper analyses would require a breakdown of flight frequencies by age which were not available to us. Cases of HCA-LD showed no apparent seasonality as they occurred almost consistently during the whole year.

Our analysis raises the following point: some travel accommodations close during winter and spring or are only partially occupied. If it is particularly the older part of the population that will use these accommodations in late spring/early summer, i.e., before the summer holidays in their own countries, there is a double risk, first because of possible stagnation issues in the accommodations and second because of the vulnerability of the tourists’ age.

Finally, the following limitations should be mentioned: as already discussed it is not easy to differentiate if travelers acquire LD during travel in a hotel or other accommodation or after they return to their own home. It is therefore not clear to which extent cases of LD are misclassified as TALD, although they are CALD. This should be investigated in future research.

Furthermore, this analysis based on general reporting data from the national surveillance system and information on exposure is not always complete.

## 5. Conclusions

Although the overall seasonality in LD is heavily influenced by CALD, seasonal differences are substantially more pronounced for TALD which may, to a large extent, reflect travel behavior. The bimodal pattern of TALD is mainly attributable to the curve among those aged 60 years and above and may reflect their preference to travel outside school holiday periods when families tend to go on vacation (August). In addition, the comparison with flight frequencies reinforces the plausibility of the course and count of TALD cases. In addition to the prevention of CALD, there is also special public health relevance for TALD. For this reason, hoteliers and owners of other accommodation sites should be aware of Legionnaire’s disease in travel accommodations and how Legionella contamination can be reduced. Especially in spring, when many accommodations reopen after seasonal winter closure, it is particularly important to take appropriate preventive measures to avoid Legionella contamination, especially since the particularly vulnerable groups travel in the pre-season in spring.

In contrast to CALD and TALD there is no apparent seasonality among HCA-LD. This reminds us that vigilance for HCA-LD needs to be high throughout the entire year.

## Figures and Tables

**Figure 1 ijerph-17-03049-f001:**
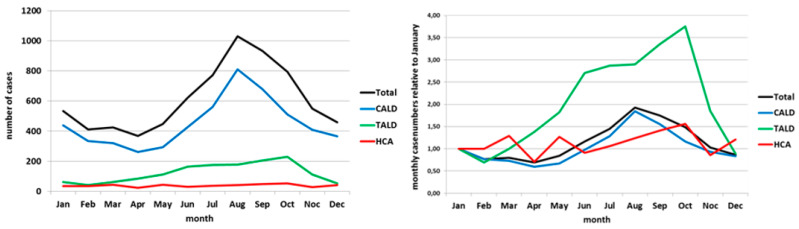
Left panel: absolute case numbers by month and exposure category; Germany, pooled data 2005–2015. Right panel: monthly case numbers (relative to January) by month and exposure category; Germany, pooled data 2005–2015.

**Figure 2 ijerph-17-03049-f002:**
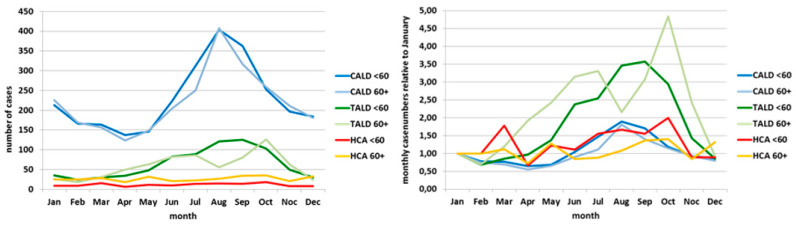
Left panel: absolute case numbers by month, age group and exposure category; Germany, pooled data 2005–2015. Right panel: relative case numbers (relative to January) by month, age group and exposure category; Germany, pooled data 2005–2015.

**Figure 3 ijerph-17-03049-f003:**
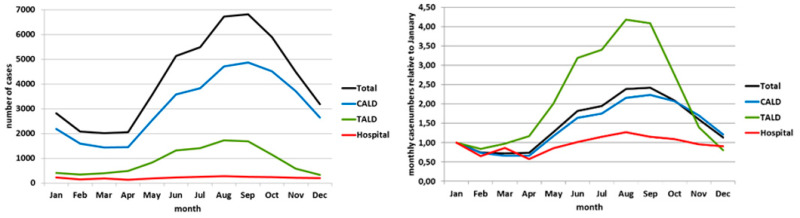
Left panel: absolute case numbers by month and exposure category, EU/EEA; European Center for Disease Prevention and Control (ECDC) “The European Surveillance System” (TESSy) data 2005–2015. Right panel: monthly case numbers (relative to January) by month and exposure category, EU/EEA; ECDC TESSy data 2005–2015.

**Figure 4 ijerph-17-03049-f004:**
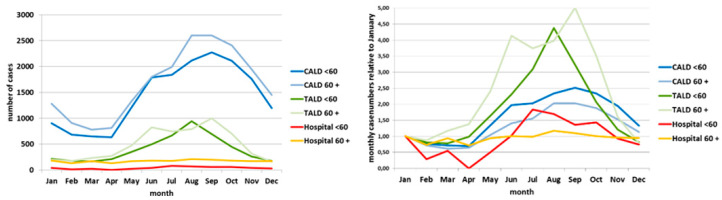
Left panel: absolute case numbers by month, age group and exposure category, EU/EEA; ECDC TESSy data 2005–2015. Right panel: monthly case numbers (relative to January) by month, age group and exposure category, EU/EEA; ECDC TESSy data 2005–2015.

**Figure 5 ijerph-17-03049-f005:**
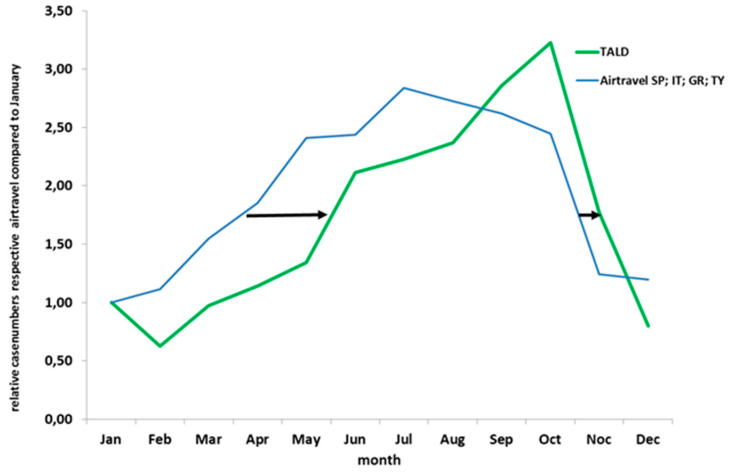
German data (pooled data 2011–2015) travel-associated cases relative to January compared to number of air travel (flights from Germany to Spain (SP), Italy (IT), Greece (GR) and Turkey (TY) in 2011–2015) by month.

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
