# Peer review of "Differential Seasonality of Legionnaires’ Disease by Exposure Category"

_ijerph, 2020, doi:10.3390/ijerph17093049_

Round 1
Reviewer 1 Report
The authors present a short study examining seasonal trends among the three categories of LD.
It is suprisingly that this analysis has not been conducted before. Although the findings pertaining to the seasonality of TALD are not entirely surprisng the data is of value to Legionella practitioners.
Given the suitable brevity of the manuscript and the clear presentation of the data there is little I can add from a reviewer perspective. However I have four specific comments:
- On page 2 lines 88-90 - text has been left from the template and should be removed.
- The authors could state more clearly what the potential public health benefits of the findings are.
- A significant portion of the travellers go abroad during the summer months when the risk of contracting LD in the destination country is naturally higher - as shown by CALD data. Couldn't this, to an extend account for the seasonality of TALD?
- The limtiations of the study/further work to do next, could be stated more clearly.
Author Response
Dear colleague,
Many thanks for reviewing our manuscript and your valuable comments and suggestions. We have taken your comments into account and adapted the document accordingly.
Please find below our answers to your suggestions:
“1. On page 2 lines 88-90 - text has been left from the template and should be removed.”
Reply: Thanks for this note; we have deleted the corresponding text.
“2. The authors could state more clearly what the potential public health benefits of the findings are.”
Reply: We´ve insert the following additional text in the discussion and conclusions: “…Our analysis raises the following point: some travel accommodations close during winter and spring or are only partially occupied. If it is particularly the older part of the population that will use these accommodations in late spring/early summer, i.e. before the summer holidays in their own countries, there is a double risk, once because of possible stagnation issues in the accommodations and second because of the vulnerability of the tourists’ age….” (see in the discussion part line 152ff)
An additional text was also added in the conclusions: “…In addition to the prevention of CALD, there is also special public health relevance for TALD. For this reason, hoteliers and owners of other accommodation sites should be aware of Legionnaire´s disease in travel accommodations and how Legionella contamination can be reduced. Especially in spring, when many accommodations reopen after seasonal winter closure, it is particularly important to take appropriate preventive measures to avoid Legionella contamination, especially since the particularly vulnerable groups travel in the pre-season in spring. …” (see line 172ff in the updated manuscript)
“3. A significant portion of the travellers go abroad during the summer months when the risk of contracting LD in the destination country is naturally higher - as shown by CALD data. Couldn't this, to an extend account for the seasonality of TALD?”
Reply: Thank you for this valuable comment. This is a good point an we´ve taken this into account and added a text in the discussion as follows: “…In addition summer tourism takes place when the risk of contracting LD (in the traveled countries) is generally higher due to higher temperatures and possibly better growth conditions for Legionella in the natural environment or cooling towers. Thus, this fact may also play a role in the seasonality of TALD…”. (see line 135ff).
“4. The limtiations of the study/further work to do next, could be stated more clearly.”
Reply: We´ve added the limitations at the end oft he discussion as follows: “….Finally the following limitations should be mentioned: As already discussed it is not easy to differentiate if travelers acquire LD during travel in a hotel or other accommodation or after return in their own home. It is therefore not clear to which extent cases of LD are misclassified as TALD, although they are CALD. This should be investigated in future research.
Furthermore, this analysis based on general reporting data from the national surveillance system and information on exposure is not always complete….” (see line 157ff)
We hope that the changes and corrections made in the manuscript are sufficient and that we were able to convince you of our manuscript.
Best regards, Bonita Brodhun
Reviewer 2 Report
A well written paper with an interesting and current topic. So far no data has been evaluated or published in this form. It was about time. Thanks a lot.
At one point or another, I would have liked to see more reference to international scientific literature - citations.
I would like to suggest two small corrections:
1. standardization of the spelling of healthcare-associated LD (HCA, line 29), abbreviated as HCA.
In line 36 it is only abbreviated with HC and in line 153, for example, HC-LD is written, which also corresponds to HCA.
2. line 74 ff "The curve of HCA cases (red line in Fig. 1 left panel) showed no apparent seasonality, whereas the curve of TALD cases had a curve suggestive of an M-like pattern (green line in Fig. 1 left panel).
There's no sense in that. In the left figure of Fig. 1, the curve of TALD looks almost like an M, but the comparison is even clearer in Fig. 2, right side, TALD 60+.
I look forward to the publication of the paper.
Author Response
Dear colleague,
Many thanks for reviewing our manuscript and your valuable comments and suggestions. We have taken your comments into account and adapted the document accordingly.
Please find below our answers to your suggestions:
“1. standardization of the spelling of healthcare-associated LD (HCA, line 29), abbreviated as HCA.
In line 36 it is only abbreviated with HC and in line 153, for example, HC-LD is written, which also corresponds to HCA.”
Reply: Thank you for this note; we have standardized the abbreviation accordingly to HCA (see in lines 36, 150 and 179/180)
“2. line 74 ff "The curve of HCA cases (red line in Fig. 1 left panel) showed no apparent seasonality, whereas the curve of TALD cases had a curve suggestive of an M-like pattern (green line in Fig. 1 left panel).
There's no sense in that. In the left figure of Fig. 1, the curve of TALD looks almost like an M, but the comparison is even clearer in Fig. 2, right side, TALD 60+.”
Reply: We´ve made little changes in the text and insert an additional sentence to clarify. “…The curve of HCA cases (red line in Fig. 1 left panel) showed no apparent seasonality, whereas the curve of CALD cases shows a clear seasonal pattern with a peak in the summer months. The curve of TALD cases rather plateaus during summer months, or even shows a dent when being “amplified” in the curve showing the relative case numbers (green line in Fig 1 right panel)…” (see line 75ff).
Of course you are right that the M-like pattern is even more obvious in figure 2 which is explained in the text some lines later (see line 82ff).
We hope that the changes and corrections made in the manuscript are sufficient and that we were able to convince you of our manuscript.
Best regards,
Bonita Brodhun